# The hydrophobic effect characterises the thermodynamic signature of amyloid fibril growth

Juami Hermine Mariama van Gils[1☯], Erik van Dijk[1☯], Alessia Peduzzo[2☯], Alexander Hofmann[3], Nicola Vettore[2], Marie P. Schützmann[2], Georg Groth[3], Halima Mouhib[4], Daniel E. Otzen[5], Alexander K. Buell[2,6]*, Sanne Abeln[1]*

**1** Computer Science Department, Center for Integrative Bioinformatics (IBIVU), VU University, Amsterdam, The Netherlands, **2** Institute of Physical Biology, University of Düsseldorf, Düsseldorf, Germany, **3** Institute of Biochemical Plant Physiology, University of Düsseldorf, Düsseldorf, Germany, **4** Universitée Paris-Est, Laboratoire Modélisation et Simulation Multi Echelle, MSME UMR 8208 CNRS, 5 bd Descartes, 77454 Marne-la-Vallée, France, **5** Interdisciplinary Nanoscience Center (iNANO) and Department of Molecular Biology, Aarhus, Denmark, **6** Department of Biotechnology and Biomedicine, Technical University of Denmark, Lyngby, Denmark

☯ These authors contributed equally to this work.
* alebu@dtu.dk (AKB); s.abeln@vu.nl (SA)

**Data Availability Statement:** The code of the model and the data and scripts used to generate the figures are available on GitHub: https://github.com/ibivu/amyloid_hydrophobicity.

## Abstract

Many proteins have the potential to aggregate into amyloid fibrils, protein polymers associated with a wide range of human disorders such as Alzheimer's and Parkinson's disease. The thermodynamic stability of amyloid fibrils, in contrast to that of folded proteins, is not well understood: the balance between entropic and enthalpic terms, including the chain entropy and the hydrophobic effect, are poorly characterised. Using a combination of theory, *in vitro* experiments, simulations of a coarse-grained protein model and meta-data analysis, we delineate the enthalpic and entropic contributions that dominate amyloid fibril elongation. Our prediction of a characteristic temperature-dependent enthalpic signature is confirmed by the performed calorimetric experiments and a meta-analysis over published data. From these results we are able to define the necessary conditions to observe cold denaturation of amyloid fibrils. Overall, we show that amyloid fibril elongation is associated with a negative heat capacity, the magnitude of which correlates closely with the hydrophobic surface area that is buried upon fibril formation, highlighting the importance of hydrophobicity for fibril stability.

## Author summary

Most proteins fold in the cell into stable, compact structures. Nevertheless, many proteins also have the ability to stick together, forming long fibrillar structures that are associated with a wide range of human disorders including Alzheimer's and Parkinson's disease. The exact nature of the amyloid-causing stickiness is not well understood, nevertheless amyloid fibrils show some very specific thermodynamic characteristics. Some fibrils even destabilise at low temperatures. In this work we translate hydrophobic theory previously

**Funding:** SA and JvG thank the Nederlandse Organisatie voor Wetenschappelijk Onderzoek (NOW, https://www.nwo.nl/over-nwo/organisatie/nwo-onderdelen/enw) for funding received under project number number 680-91-112. The simulations described in this work were carried out on the Dutch national e-infrastructure with the support of SURF Cooperative (EvD and SA, SH-309-14, https://userinfo.surfsara.nl/systems/cartesius). AKB thanks the European Molecular Biology Organization (EMBO, grant number ASTF 242.00-2011, https://www.embo.org/funding-awards/fellowships/short-term-fellowships), Magdalene College, Cambridge for funding (no grant number, https://www.magd.cam.ac.uk/), and the Parkinson's and Movement Disorder Foundation (PMDF, no grant number, https://www.pmdf.org/) for funding and the Novo Nordisk Foundation for support through a Novo Nordisk Foundation Professorship (NNFSA170028392). HM thanks the Centre national de la recherche scientifique (CNRS) InFinity program (no grant number, http://www.cnrs.fr/) for funding and RWTH Aachen University for a Theodore von Kármán Fellowship (no grant number, https://www.rwth-aachen.de/cms/root/Forschung/Angebote-fuer-Forschende/ERS-Angebote/ERS-International/~rohj/Theodore-von-Karman-Fellowship-Outgoin/). DEO acknowledges support from the Lundbeck Foundation (grant R276-2018-671, https://www.lundbeckfonden.com/en/). The funders had no role in study design, data collection and analysis, decision to publish, or preparation of the manuscript.

**Competing interests:** The authors have declared that no competing interests exist.

used to model protein folding to fibril formation. We combine this theory with experimental measurements, simulations and meta-data analysis of different types of fibrils. This allowed us to unravel the nature of the stickiness in amyloid fibrils by observing the effect of temperature changes, specifically at low temperatures, on hydrophobicity.

## Introduction

The folding of proteins is an essential process for cellular functioning. Protein misfolding and aggregation on the other hand can severely deregulate cells. The most dominant energetic contributions to protein folding include pairwise amino acid interactions, the hydrophobic effect and the configurational entropy of the polypeptide chain; these contributions can be captured by a range of models describing the process at different temperatures: [1–13]. The protein folding process involves several enthalpy-entropy compensating mechanisms. The hydrophobic effect, the main stabilizing factor of the folded state [14], contains both enthalpic and entropic components [15–17]; this is apparent from the weakening of this effect at low temperatures [18]. Similarly, the chain configurational entropy and the pairwise amino acids interaction can lead to enthalpy-entropy compensation; this is apparent from the heat-induced unfolding of proteins [1–6]. The accurate description of the hydrophobic effect within several models for protein folding, using both the enthalpic and entropic terms, has enabled the reproduction of experimentally observed features, such as cold-, heat-, and pressure-induced denaturation [9, 19–27], thereby emphasizing also the role of hydrophobicity in the stability of folded proteins (see Fig 1A for an example of a folded protein with a hydrophobic core). Thus, the enthalpy-entropy compensation mechanisms in protein folding are well-understood and important for rationalising the thermodynamic characteristics of protein folding.

Under certain conditions, specific proteins can aggregate into $\beta$-strand-dominated amyloid fibrils. This process is believed to be the underlying cause of many degenerative diseases, such as the aggregation of the A$\beta$ peptide in the case of Alzheimer's disease and of $\alpha$-synuclein in the case of Parkinson's disease [29]. Unlike protein folding, the fibril formation process is not well-understood in terms of its thermodynamic characteristics.

Recently resolved full length fibril structures, e.g. [28, 30], suggest that the cores of disease-associated amyloid fibrils are characterised by a higher average hydrophobicity than the overall sequences, see Fig 1C, indicating that hydrophobicity may play an important role in the formation of amyloid fibrils. The thermodynamic properties of amyloid fibrils at different temperatures can be investigated experimentally, for example by using Differential Scanning Calorimetry (DSC, [31, 32]) or Isothermal Titration Calorimetry (ITC) [33–35]. Under constant pressure, the heat ($dQ$) added or removed from the system is equal to its change in enthalpy ($dH$). ITC experiments show that amyloid fibril growth is generally an exothermic process [33, 34] with a negative heat capacity [34, 35]. However, a few notable exceptions with positive heat capacity have been reported [34, 35]. Under physiological conditions, amyloid fibrils are often very stable [36], but monomer solutions can nevertheless be metastable for extended amounts of time, as high free energy barriers can prevent aggregates from forming [37]. Amyloid fibrils are often relatively thermostable, but will eventually heat denature at sufficiently high temperatures [31, 32]. Interestingly, a small number of fibril systems has been reported to also denature at temperatures near the freezing point [35, 38]. In contrast to protein folding, it is currently unclear how the different enthalpic and entropic contributions to amyloid stability may lead to these temperature dependent effects, and how the experimental thermodynamic signatures of fibril elongation may be rationalised. In order to elucidate the

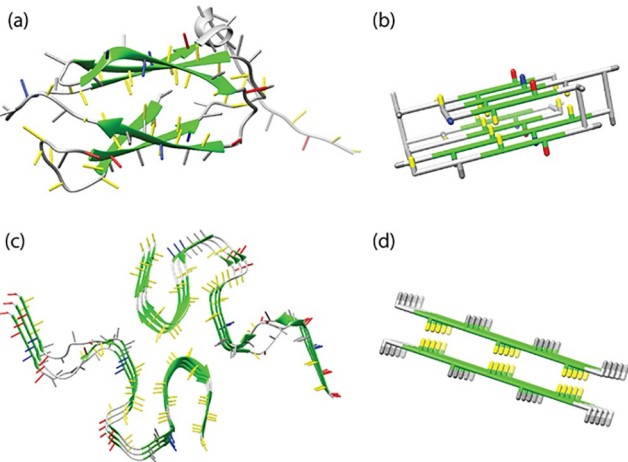

**Fig 1. Folded and fibrillar states.** (A) The native state of human transglutaminase (PDB-ID: 2XZZ) (B) a model protein in the native state, (C) disease-related amyloid $\beta$-sheet of the A$\beta$ (1-42) peptide (PDB-ID: 2NAO) [28] and (D) the seed structure of an amyloid fibril represented in the lattice model used for this work. Side chains of hydrophobic residues are coloured in yellow, of polar residues in grey, of positively charged residues in blue and of negatively charged residues in red. For $\beta$-stranded structures the backbone is coloured in green. In the folded protein, a hydrophobic core can be observed, where the hydrophobic residues are shielded from the water by the hydrophilic and charged residues. Similarly, the core sequence regions of amyloid fibril structures tend to be strongly hydrophobic.

molecular origins of the stability of amyloid fibrils and the associated thermodynamic signatures, insight from theoretical and computational models is required.

Here, we use a combination of theory, simulations, experiments and meta-data analysis to determine the entropic and enthalpic contributions of amyloid fibril elongation.

For the simulations, we used the Monte Carlo method on a coarse-grained physical protein model situated on a cubic lattice to study amyloid fibril elongation. Previously, we included the temperature dependence of the hydrophobic effect in a lattice model to delineate the enthalpic and entropic contributions of protein folding [7, 9]; however, this classic lattice model is unsuitable to study amyloid fibril formation. We also developed a different model that includes hydrogen-bond dependent beta-strand formation [39] to simulate amyloid fibril formation [39–43]. The coarse-grained lattice model used in this work therefore incorporates elements of these two previous models: hydrogen-bond dependent beta-strand formation [39] and the temperature dependence of the hydrophobic effect [9]. It explicitly captures the dominating enthalpic and entropic components, including: chain entropy, entropy-enthalpy compensation from the hydrophobic effect, and enthalpic terms from hydrogen bonds and side-chain interactions. Fig 1B shows a folded protein represented by the model.

In this study, we aim to investigate the thermodynamic signatures and the temperature dependence of amyloid fibril formation. We focus on the *elongation* step in amyloid fibril formation process, as shown in Fig 1D. Using this approach, both in the simulations and experiments, we do not have to consider the kinetic or thermodynamic characteristics of the nucleation processes and the possible formation of oligomers as intermediate states before the formation of well-defined fibrils. The focus on the elongation step allows us to simplify the simulation setup, and to compare our simulations directly with microcalorimetry and equilibrium data from the literature and from our own experiments. We have gathered an extensive collection of thermodynamic data of peptide assembly and we find that the predictions of our model are corroborated by these results.

Using this combined computational and experimental approach, we show that cold denaturation only occurs when (1) the hydrophobic effect represents a significant component of the fibril stability, (2) the overall stability of the fibril is sufficiently low and (3) fibril elongation changes from being an endothermic reaction at lower temperatures to an exothermic reaction at higher temperatures. Furthermore, from our experiments and meta-data analysis we can conclude that the strength of the temperature effect on the enthalpy of fibril elongation (i.e. the absolute value of the heat capacity) is directly dependent on the hydrophobicity of the sequence region of the polypeptide that forms the core of the fibril.

## Materials and methods

This work consists of four main pillars: (I) Theory describing the temperature dependence of the hydrophobic effect derived from previous works [9], (II) Monte Carlo simulations of a lattice model that captures the fibril elongation process, (III) *in vitro* measurements of the enthalpy and free energy of fibril elongation for different fibril systems and (IV) meta-data analysis of the relation between the enthalpy of fibril elongation and the buried hydrophobic surface area for fibrils from different peptides and proteins (see Fig 2).

### Theoretical temperature dependence of the hydrophobic effect

In this work, we define a temperature dependent solvation term $\Phi_{\text{solvent}}(T)$ that can account for the temperature dependence of the hydrophobic effect.

$$\Phi_{\text{solvent}}(T) = \sum_i^N F_{\text{hydr}}(i) + \epsilon_{a_i,\text{solv}} K_{i,\text{solv}} \tag{1}$$

Here $N$ is the total number of residues in a peptide; the term $\epsilon_{a_i,\text{solv}}$ is an interaction energy between the solvent and the residue that does not depend on the temperature, whereas the free energy term, $F_{\text{hydr}}(i)$, does. $K_{i,\text{solv}}$ indicates whether or not an interaction between the solvent and residue $i$ occurs. Previously, we used a very similar solvation term to model the temperature dependence of the hydrophobic effect in protein folding [9].

We can describe the hydrophobic temperature dependence, $F_{\text{hydr}}(i)$, for a residue $i$ as:

$$F_{\text{hydr}}(i) = -\alpha C_h (T - T_0)^2 \tag{2}$$

where $T$ is the temperature, $T_0$ sets the temperature at which the hydrophobic effect is maximal, $\alpha$ defines the strength of the hydrophobic temperature dependence and $C_h$ indicates whether a hydrophobic residue makes contact with the solvent.

### Delineation of enthalpic and entropic hydrophobic contributions

From the hydrophobic free energy contribution, as given by Eq 2, we can also deduce the enthalpic contribution. Multiplying both sides by $\beta \left( = \frac{1}{T} \right)$, taking derivative with respect to $\beta$ and using that $\frac{d\beta F}{d\beta} = \langle E \rangle$ we obtain:

$$\langle E_{\text{hydr}}(i) \rangle = -\alpha C_h (T_0^2 - T^2) \tag{3}$$

This equation can be used to estimate the difference in hydrophobic enthalpy between two states, based on the difference in hydrophobic residues pointing towards the solvent (see S1

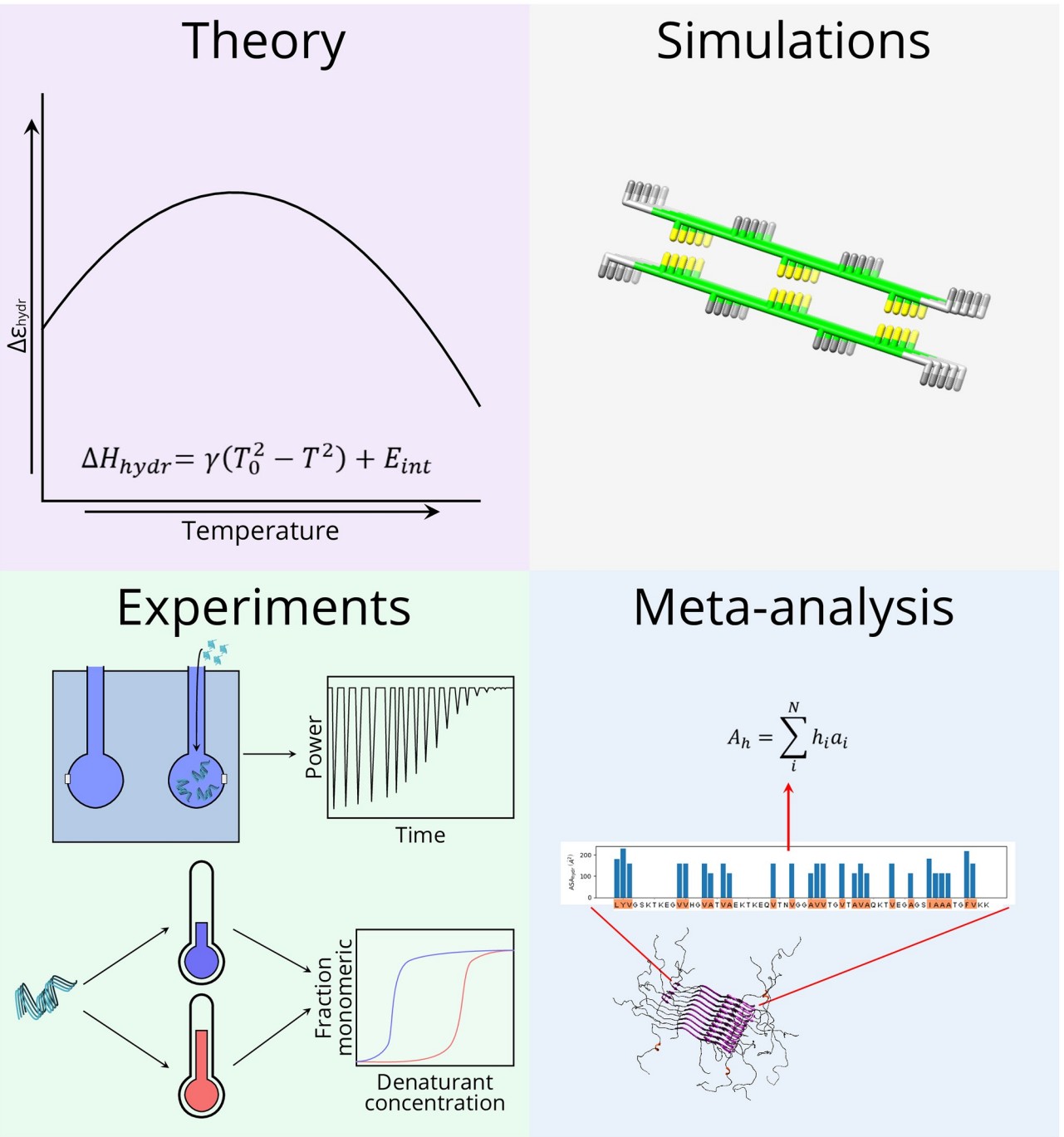

**Fig 2. Main techniques used in this work.** This work contains four main components: (I) Theory developed from previously published results, (II) A lattice model of fibril elongation, (III) Experimental measurements of the free energy and enthalpy of amyloid fibril elongation and the enthalpy of short peptide assembly, (IV) Meta-data analysis of the enthalpy of fibril elongation.

Methods). Note that we can now also compute the entropic contribution using:

$$-TS_{\mathrm{hydr}}(i) = F_{\mathrm{hydr}}(i) - E_{\mathrm{hydr}}(i) \tag{4}$$

## Theoretical estimates for hydrophobic contributions

We also try to derive theoretical estimates of the hydrophobic contributions to the free energy, entropy and enthalpy of fibril elongation.

If we consider only the contributions that dominate the free energy differences between states, the free energy difference of fibril elongation can be approximated by:

$$\Delta G = \Delta G_{\mathrm{hydr}} + \Delta G_{\mathrm{int}} + \Delta G_{\mathrm{chain}} \tag{5}$$

We can simplify this further in our model by noting that the interaction energies, including the backbone hydrogen bonds, do not have a temperature dependence. Moreover, the main contribution of the chain free energy is entropic. Note that this component will turn out to be strongly temperature dependent in our simulations.

$$\Delta G \cong \Delta G_{\mathrm{hydr}} + \Delta E_{\mathrm{int}} - T\Delta S_{\mathrm{chain}} \tag{6}$$

In order to make a theoretical estimate, we can consider the case where the hydrophobicity dominates the change in free energy upon fibril elongation. Then, we can calculate an estimate for the change in free energy, $\Delta\hat{G}_{\mathrm{hydr}}$, as:

$$\Delta\hat{G}_{\mathrm{hydr}} = \Delta G_{\mathrm{hydr}}(T) + \Delta E_{\mathrm{int}} \tag{7}$$

We can then estimate the change in enthalpy as:

$$\Delta\hat{E}_{\mathrm{hydr}} = \Delta E_{\mathrm{hydr}}(T) + \Delta E_{\mathrm{int}} \tag{8}$$

From these estimates we can calculate the entropic contribution as:

$$-T\Delta\hat{S}_{\mathrm{hydr}} = \Delta\hat{G}_{\mathrm{hydr}}(T) - \Delta\hat{E}_{\mathrm{hydr}} \tag{9}$$

Note that we will later show that only the enthalpic signature derived by these estimates will agree with the actual simulation results.

## The peptide simulation model

Experimentally, it has been shown that amyloid formation is usually strongly accelerated in the presence of preformed fibrillar aggregates, or seeds [44]. In this study, we focus on the thermodynamic properties of elongation, or the addition of a single protein molecule to the end of a 'seed' fibril [45]. We study this process using a cubic lattice model, where each amino acid occupies a single grid site. In this model, inspired by previous work on protein lattice models [1, 3–5, 46, 47], each amino acid interacts with the amino acids directly adjacent to it. If no amino acid is present, the grid site is assumed to be occupied by the solvent [7].

The side chain is modelled by giving each amino acid an orientation and allowing hydrogen bonds to be formed only when the side chains point in the same direction, allowing a reasonable steric approximation of $\beta$-strands [39]. For the peptides we used the same sequences as in ref. [39]: TFTFTFT. To investigate the effect of a lower stability, we replaced the phenylalanine residues with leucine residues, yielding the sequence TLTLTLT. The default parameters for the model are comparable to settings that are required to obtain self-replication in a simpler model [11].

In our simulations, a seed for the study of the elongation process is represented by a pre-formed fibril consisting of 8 peptide molecules. This fibril is 'frozen' during our simulations, meaning that only moves are allowed that translate or rotate the entire seed fibril. However, all interactions of the fibril with the environment are still present. Two additional monomeric protein molecules are present in the simulation box, which are allowed to make regular moves, and can attach and detach from the seed during the simulation. When both molecules are in the monomeric state, this leads to a monomer concentration of 4.6 mM. This setup allows us to investigate the addition of one layer (consisting of two molecules) to the pre-formed seed fibril. In the simulations, we observe four distinct states: monomeric, amorphous, fibrillar and fully aggregated. We define these states based on the total number of external contacts (see the S1 Methods).

A full description of the model is given by the following equation:

$$\mathcal{H} = E_{\mathrm{hb}} + E_{\mathrm{steric}} + E_{\mathrm{state}} + E_{\mathrm{aa}} + \Phi_{\mathrm{solvent}}(T) \tag{10}$$

This Hamiltonian, $\mathcal{H}$, is given in reduced units ($k_B T$ units). The term $E_{\mathrm{aa}}$ represents the sum of the classical pairwise amino acid interactions $\epsilon_{a_i,a_j}$, that are used in most coarse-grained simulations. The term $E_{\mathrm{hb}}$ represents the interactions originating from hydrogen bonds between side chains of two amino acids. $E_{steric}$ represents the steric interactions. $E_{\mathrm{state}}$ represents the energy gained from $\beta$-sheet formation. $\Phi_{solvent}(T)$ is the interaction of an amino acid with the solvent. The first four terms are kept identical to the model described by Abeln et al. [39]. Note that only $\Phi_{solvent}(T)$ depends on the temperature, whereas the other four terms do not. Hence, we can also describe the Hamiltonian in terms of a temperature-independent and a temperature-dependent term $\mathcal{H} = E_{int} + \Phi_{solvent}(T)$.

**Hydrophobic temperature dependence in the simulation model.** Eqs 1 and 2 define how the hydrophobic temperature dependence is included in our simulation model. Previously, we used a very similar solvation term to include the temperature dependence of the hydrophobic effect in protein folding simulations. There we used two different variants: a single and a two-state model for the solvation term of a side-chain with the solvent. Both variants gave similar results for the thermodynamic signatures of protein folding [9]. Here we use the simplest formulation of the model—with fewer parameters—that is based on a single-state representation for a side-chain with the solvent.

In our simulations, $K_{i,\mathrm{solv}} = 1$ when the side chain points in the direction of the solvent, and $K_{i,\mathrm{solv}} = 0$ otherwise. $C_h$ calculated as $h_{a_i} K_{i,\mathrm{solv}}$, where $h_{a_i}$ indicates if the amino acid is hydrophobic ($\epsilon_{a_i,w} > 0$). In our potential, the amino acids that fulfil this condition are $a_i \in \{C, F, L, W, V, I, M, Y, A\}$.

**Estimate for hydrophobic enthalpy in the simulation model.** By combining Eqs 3 and 8 we can obtain a relation from which we can estimate the full enthalpic contribution due to the hydrophobic effect:

$$\langle \hat{E}_{hydr}(i) \rangle = \gamma(T_0^2 - T^2) + E_{int} \tag{11}$$

where $\langle \hat{E}_{hydr}(i) \rangle$ is an estimate for the hydrophobic enthalpy, $\gamma$ is the strength of the hydrophobic effect, $T_0 = 0.4$ (reduced units) is the optimal temperature for hydrophobic interactions, $T$ is the temperature and $E_{int}$ is the internal energy of the protein. For the simulation model we have $\gamma = \alpha C_h$. Now by taking $\Delta C_h = 6$ for the difference in solvent contacts between the fibrillar and monomeric state, and taking $E_{\mathrm{int}}$ from the constant term observed in the simulations with $\alpha = 0$, we can make the estimates for the hydrophobic contributions described above.

Further details on the simulation model, including descriptions of the different fibrillar states, sampling procedures and the investigation of entropically favourable $\beta$-sheets, can be found in S1 Methods [48–50].

### *In vitro* experiments

**Preparation of (seed) fibrils.**   $\alpha$-synuclein (wild type and F94W mutant) was recombinantly expressed and purified as reported previously [44]. Solutions of $\sim 200$ $\mu$M $\alpha$-synuclein in phosphate buffer saline (PBS) and 0.02% NaN$_3$ were prepared and incubated under strong stirring at temperatures between 30-37˚C for several days. An AFM image of the amyloid fibrils obtained in this manner is shown in S5A Fig. Bovine $\alpha$-lactalbumin was purchased from Sigma and used without further purification. Solutions of 200-350 $\mu$M of $\alpha$-lactalbumin in 10 mM HCl and 100 mM NaCl were incubated at 37˚C under constant stirring for 2-3 days. Human glucagon was a kind gift from Novo Nordisk. Glucagon was studied under two different sets of solution conditions. Solutions were prepared at $\sim 700$ $\mu$M in 10 mM HCl and 1 mM Na$_2$SO$_4$ and incubated under stirring at 40˚C for 2 days. Alternatively, solutions were prepared at $\sim 300$ $\mu$M in 10 mM HCl and 30 mM NaCl and incubated under stirring at 40˚C for 1 day.

**Isothermal titration calorimetry (ITC) experiments of fibril elongation.**   We performed ITC experiments to directly probe the enthalpy change associated with the addition of a protein monomer to the end of an amyloid fibril, as a function of temperature [33, 34]. An ITC experiment of fibril elongation can be carried out in two distinct ways, by injecting seed fibrils into monomeric protein solutions, or by injecting monomeric protein into seed fibril suspensions. In the present study, we have explored both types of experiments. The injection of monomeric protein into seed fibrils can be carried out repeatedly, whereas a single injection of seed fibrils into a supersaturated solution of monomers leads ultimately to a complete conversion of the solution into aggregates. The ITC experiments of fibril elongation were carried out with VP-ITC and ITC200 instruments (Malvern Instruments, UK). In the case of $\alpha$-synuclein (VP-ITC and ITC200), both types of experiments (seed fibrils into monomer and monomer into seed fibrils) were performed, whereas in the case of glucagon (VP-ITC) and bovine $\alpha$-lactalbumin (VP-ITC), monomer titrations into fibril suspensions were used. In the case of glucagon and $\alpha$-lactalbumin, the monomer solutions and the fibril solutions were dialysed overnight at 4˚C against a large volume of the same solution in order to ensure that the heat released or consumed upon titration only corresponds to the heat of reaction and not potential heats of dilution of unbalanced salt concentrations. The seed fibrils were sonicated with a probe sonicator according to protocols similar to the ones reported in [51] in order to maximise the seeding efficiency and accelerate the reaction. A crucial point in these experiments is that the rate of heat release or consumption is sufficiently high to produce a clearly visible peak that can be integrated to yield the total amount of heat exchanged due to fibril elongation. At higher temperatures, more heat is released (due to the strongly negative heat capacity of the elongation reaction, see Results) and the rate of fibril elongation is accelerated [37], and both of these factors are beneficial for the signal-to-noise ratio of the experiment. In order to be able to perform reliable measurements at the lower end of the temperature range investigated, the only possibility to accelerate the reaction is by increasing the number of growth competent fibril ends. S5B Fig shows an AFM image of sonicated $\alpha$-synuclein amyloid fibrils. The ITC experiments of fibril elongation were performed at monomer concentrations between 100-350 $\mu$M ($\alpha$-lactalbumin), between 70-300 $\mu$M (glucagon) and between 50-100 $\mu$M ($\alpha$-synuclein). S5C Fig shows an AFM image of $\alpha$-synuclein amyloid fibrils taken out of an ITC calorimeter after an experiment.

The heat released or consumed upon an injection of monomer or fibrils, $\Delta Q$, was determined and divided by the amount of monomer that had reacted in each case. S4 Fig shows raw data from ITC experiments of amyloid fibril elongation for the proteins $\alpha$-lactalbumin, $\alpha$-synuclein and glucagon.

Furthermore, we use $\Delta E$ and $C_v$ when we refer to the simulations, since they are carried out at constant volume, and $\Delta H$ and $C_p$ when we refer to the experiments, since they are carried out at constant pressure. As described in [9], these are very similar in our setup. Finally, we use $\Delta F$ to refer to the free energy defined in the model, and $\Delta G$ when we refer to free energy sampled by the simulations, as in [9].

Additionally, ITC experiments were performed on GNNQQNY, a small fragment of the Sup35 protein that can form amyloid-like microcrystals. See the S2 Methods for additional experimental details [52].

**Fibril stability in the presence of denaturants.**   We have determined the thermodynamic stabilities of amyloid fibrils using depolymerisation experiments with a chemical denaturant, using the isodesmic linear polymerisation model, as previously described [36, 53]. We have recently presented an extension of the standard framework for the analysis of such chemical depolymerisation of amyloid fibrils, which captures better the dependence on protein concentration [54]. However, here we analyse our data with the simplified isodesmic model, in order to achieve easy comparability with the free energy values of amyloid fibril growth reported previously [36]:

$$y = \frac{2[M]_T e^{-\frac{(\Delta G^0 + m[D])}{RT}} + 1 - \sqrt{4[M]_T e^{-\frac{(\Delta G^0 + m[D])}{RT}} + 1}}{2[M]_T{}^2 \left( e^{-\frac{(\Delta G^0 + m[D])}{RT}} \right)^2} \tag{12}$$

where y is the fraction of monomeric protein, $[M]_T$ is the total protein concentration, $\Delta G^0$ is the free energy change upon addition of a monomer to the fibril, R is the gas constant, T is the temperature, and m describes the sensitivity of the free energy on the denaturant concentration [D].

These denaturation experiments were performed on $\alpha$-synuclein mutant F94W, glucagon and $\alpha$-lactalbumin (see Fig 6 and S6 Fig). We have recently characterised in detail and validated the use of intrinsic Trp fluorescence as a powerful tool to determine the intrinsic thermodynamic stability of amyloid fibrils [54]. In the case of $\alpha$-synuclein, we also performed some cold denaturation experiments with the WT sequence (which does not contain Trp residues), using Thioflavin-T fluorescence and CD spectroscopy, in order to validate the results obtained with the F94W mutant (see S10 Fig).

We used different denaturants for each of the proteins. $\alpha$-synuclein (WT and F94W) was depolymerised with urea, a neutral denaturant, which allows to probe the contribution of electrostatic interaction on amyloid fibril stability. Glucagon fibrils were depolymerised with GndHCl, as the use of urea requires a certain buffer capacity of the solution [54], which was not possible, as we wanted to perform the equilibrium experiments under the same solution conditions as the calorimetry experiments. $\alpha$-lactalbumin fibrils were depolymerised with GndSCN, which also necessitated the direct measurement of the soluble protein (Bradford reagent), as the strongly absorbing thiocyanate ion prevented the use of intrinsic fluorescence as in the cases of glucagon [54] and $\alpha$-synuclein. See the S2 Methods for additional experimental details.

## Meta-data analysis

**Estimating the strength of the hydrophobic effect in amyloid fibrils.** To determine the temperature dependence of the relationship between the strength of the hydrophobic effect and the hydrophobicity of a protein, we combined our peptide model with experimental data from isothermal titration calorimetry (ITC) of amyloid fibril growth and temperature-dependent equilibrium experiments of peptide assembly into crystals. We used a least-squares fit of the temperature dependence of the enthalpy of assembly to Eq 11, with $T_0 = 343.15K$. Note that for a fully atomistic representation we have $\gamma = -\alpha A_h$, where $\alpha$ is the strength of the hydrophobic effect per unit of hydrophobic surface area and $A_h$ is the total aggregating hydrophobic surface area of the protein. To calculate $A_h$, we used

$$A_h = \sum_i^N h_i a_i \tag{13}$$

where $h_i$ indicates whether or not the amino acid at position $i$ is hydrophobic and $a_i$ indicates the surface area of the amino acid at position $i$. In this case,

$$h_i = \left\{ \begin{array}{ll} 1 & \text{if } h_i \in \{C, F, L, W, V, I, M, Y, A\} \\ 0 & \text{otherwise} \end{array} \right\}$$

For the calculation of the change in hydrophobic surface area upon addition to the fibril end, we only included the contributions of those sequence regions that are part of the $\beta$-sheet core of the amyloid fibril. These regions were determined using sources from [30, 55–61]. For a detailed description of how these core regions were determined, see the S3 Methods.

**Enthalpy of (de)solvation of small hydrophobic amino acids.** In order to cover a wider range of total hydrophobic surface area, we also included thermodynamic data of the self-assembly of a hydrophobic amino acid (phenylalanine [60]) and of a hydrophobic dipeptide (diphenylalanine [61]). The temperature dependent enthalpy of dissolution of solid phenylalanine, the reverse of the enthalpy of assembly, was measured calorimetrically [60], whereas for diphenylalanine, a van't Hoff analysis of the solubility was performed [61]. In both cases, the entire hydrophobic surface area was assumed to become buried upon assembly, given that both components assemble into crystals, rather than amyloid fibrils.

## Results and discussion

In this work, we aim to investigate the effect of interplay between the hydrophobic temperature dependence, configurational chain entropy and pairwise enthalpic interactions between amino acids (including hydrogen bonds) on the thermodynamic characteristics of amyloid fibril elongation. To achieve this, we use a combination of theory, *in vitro* experiments, computer simulations and meta-data analysis.

## Theoretical model

First, we consider if the addition of the hydrophobic effect could lead to a destabilisation of an amyloid fibril at low temperatures and ultimately to cold denaturation.

We describe the free energy contribution of the hydrophobic effect as

$$F_{hydr} = -\alpha C_h (T - T_0)^2$$

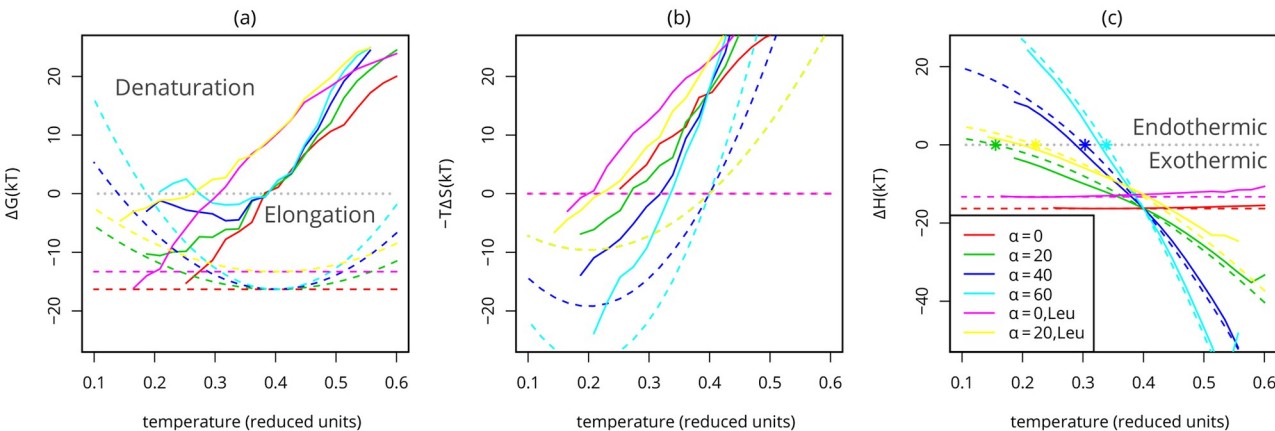

**Fig 3. Temperature dependence of the free energy components.** (A) Free energy, (B) Entropy and (C) Enthalpy of fibril elongation. The differences between the fully formed fibril ($C_{ext} = 21$) and the monomeric state ($C_{ext} = 0$) are calculated from the simulation of several different fibrils. The interaction strength of the Leucine-based fibrils is weaker, leading to a lower enthalpic contribution, effectively weakening the fibrils. Dotted lines indicate estimates for the hydrophobic contributions showing from left to right $\Delta\hat{G}_{hydr}$ and $\Delta\hat{E}_{hydr}$; these estimates are generated using Eqs 7 and 8 with corresponding $\alpha$, $\Delta C_h = -6$ and with an offset, $E_{int} = \Delta H$ based on simulations with the equivalent peptide for $\alpha = 0$; stars indicate the change of an exothermic to an endothermic process, based on the $\Delta\hat{E}_{hydr}$ estimate. It is clear that in our model the slope of the enthalpy of fibril elongation as a function of temperature, is dominated by the temperature dependence of the hydrophobic effect.

from which we can derive the enthalpy $\langle E \rangle = \frac{d\beta F}{d\beta}$, with $\beta = \frac{1}{T}$:

$$\langle E_{hydr} \rangle = -\alpha C_h (T_0^2 - T^2)$$

where $\alpha$ is the strength of the hydrophobic effect, $C_h$ is the number of hydrophobic contacts (in simulations; this term is replaced with the buried hydrophobic surface area ($A_h$) for real fibrils), $T$ is the temperature and $T_0$ is the temperature with the maximum strength of the hydrophobic effect. From the free energy and enthalpy, we can also derive the hydrophobic contribution to the entropy as $-T\Delta S_{hydr} = \Delta F_{hydr} - \Delta E_{hydr}$. This hydrophobic term has successfully been applied to account for cold denaturation of folded proteins [9]. The theoretical contributions of the hydrophobic term to the different free energy components for different strengths of the hydrophobic effect ($\alpha$) is shown in the dotted lines in Fig 3. We compare these theoretical contributions to Monte Carlo simulations of a lattice model that includes both the hydrophobic term as well as other types of interactions such as hydrogen bonds and steric interactions that are normally found in proteins (see solid lines in Fig 3) and experimental measurements of the enthalpy of fibril elongation.

## Coarse-grained simulations of fibril elongation

Here, we use a simple model of a short (seed) fibril that is made up of two layers of peptides with alternating hydrophobic and hydrophilic amino acids [39]. The fibril structure has a hydrophobic core between the two layers, as shown in Fig 1D. We sample this model by considering the process of fibril elongation: two peptides of the fibril are free to restructure in the simulations, while the remaining peptides in the seed fibril remain structurally fixed (Fig 4). Finally, we add an explicit temperature dependence for hydrophobicity [9]. In simulations of our model of peptide aggregation, all fibrils denature into monomers at high temperatures, as shown in Fig 4 and S1 Fig, consistent with our expectations of this model [39]. Experimentally, heat denaturation has also been shown for several types of fibrils [31, 35] and is even used for the destruction of contaminating amyloid fibrils, such as prions, in a clinical context [62].

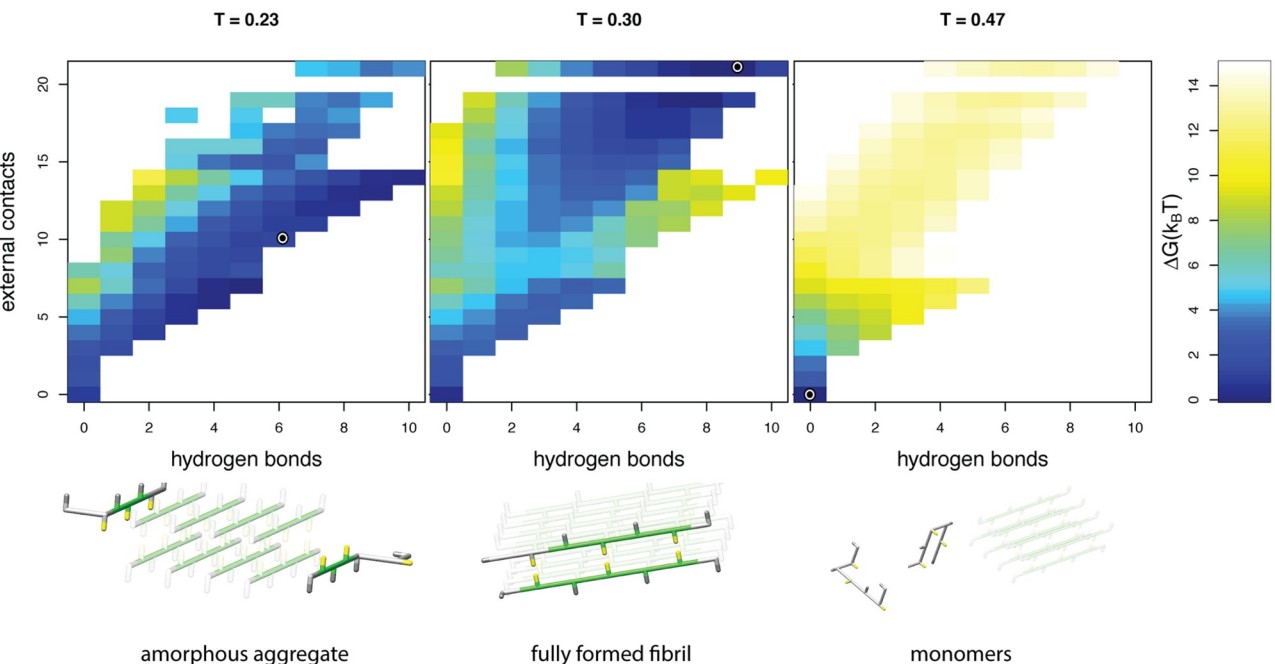

**Fig 4. Temperature dependent states.** Top: free energy landscapes, and bottom: representative snapshots are shown for simulations at three different temperatures modelled with a strong hydrophobic temperature dependence, $\alpha = 60$. The snapshots of the peptides are representative for the low free energy conformations; the circle represents the exact order parameters for the snapshot in the corresponding free energy landscape. Within the snapshots the seed fibril is indicated in a lighter shade (see caption Fig 1 for colour coding). At high temperatures only transient (external) contacts are formed and the protein molecules that are not part of the seed remain in their monomeric form. At intermediate temperatures the free peptides adopt a regular, fibrillar structure at the end of the seed. At low temperatures the heatmap shows that there is not a single distinct conformation with a low free energy: the simulated free peptides are attached to the seed fibril, but no hydrophobic core is formed between the peptides.

## Modeling cold destabilisation and denaturation

To investigate the effect of the hydrophobic temperature dependence, we vary a parameter $\alpha$ that sets the strength of the hydrophobic effect, see Eq 2 in the Materials and methods; a similar parameter was used in earlier work to model the cold denaturation of folded proteins [9]. Destabilisation of the fibril at lower temperatures is only observed when a moderately strong temperature dependence of the hydrophobic effect is included in the model ($\alpha = 40$ and $\alpha = 60$). In our simulations with $\alpha = 60$, we observe heat denaturation, as well as cold-destabilisation and even cold denaturation of the fibril into a less ordered aggregated ('amorphous'), as well as into the monomeric state ($\alpha = 60$, $T \approx 0.25$), see S1 Fig.

When we investigated the aggregated and denatured states in more detail (Fig 4), we found that the denatured state at low temperatures is structurally distinct from the heat denatured state: the low temperature amorphous structural ensemble displays more residual interactions (left panel Fig 4). This is consistent with experimental results showing that under some conditions, oligomeric states can be found at low temperatures [38], as opposed to the monomeric protein molecules that are found at very high temperatures [31, 35]. The more compact configurational ensemble of the cold denatured state of a fibril shows clear parallels with cold denaturation of monomeric proteins [9, 63].

## Computational thermodynamic signature of cold denaturation

To shed more light on the nature of the cold denaturation transition, we consider the free energy difference between the fibrillar and monomeric states. When the free energy of the

monomeric, or denatured state becomes lower than the fibrillar state, the fibril will dissolve; this corresponds to a $\Delta G > 0$ in Fig 3A. It is important to note here that this definition of the free energy in the simulation prevents a direct comparison with the free energies determined in the equilibrium depolymerisation experiments described below. In the simulations, a value of $\Delta G = 0$ corresponds to a monomer ('critical') concentration of 4.6 mM, whereas the experimental free energy scale is such that $\Delta G = 0$ corresponds to a nominal critical concentration approximately two orders of magnitude higher and whose precise value depends on the total peptide concentration. Therefore, there is a difference of the order of 5 $k_B$T between the computational and experimental free energy scales.

When we consider the enthalpic and entropic contributions of fibril stability separately, in Fig 3C and 3B respectively, it is clear that these individual contributions are large compared to the overall difference in free energy. Similar observations on enthalpy-entropy compensation have for example been made for both the stability of folded proteins [64], and the free energy barriers of amyloid fibril growth [37].

Moreover, comparing the different contributions to the free energy of elongation it is clear that the configurational chain and diffusional entropy that is lost upon binding dominates the free energy difference at high temperatures, causing the heat denaturation of the fibril. This can be most clearly seen comparing Fig 3A and 3B for $\alpha = 0$.

For cold denaturation on the other hand, it is the hydrophobic temperature dependence that dominates this transition (cyan curves in Fig 3A and 3C). Note that free energy and entropy components still contain rather large configurational contributions at these low temperatures; they do not precisely follow the hydrophobic components.

Nevertheless, the enthalpy of fibril growth strongly correlates with the signature of the hydrophobic effect ($\Delta \hat{E}_{hydr}$), as seen in Fig 3C. The temperature independent potential ($\alpha = 0$) leads to an approximately constant enthalpy of fibril elongation. Adjusting the strength of the hydrophobic effect through a change of $\alpha$ leads to a negative slope in the enthalpy as a function of temperature, which corresponds to a negative heat capacity of the elongation reaction, since $\frac{\partial \Delta E_{el}}{\partial T} = \Delta C_{p,el}$. Hence, the temperature dependence of the hydrophobic effect can be probed by measuring the steepness of the negative slope of the enthalpy of fibril elongation.

## Experimental thermodynamic signature of cold denaturation

In our simulation model, the inclusion of the hydrophobic temperature dependence effectively yields a negative value for heat capacity of the elongation reaction, where we assume $\Delta C_{v,el} \approx \Delta C_{p,el}$ as in Ref. [9].

A negative value of $\Delta C_{p,el}$ is found for most amyloid fibrils in experimental work [33–35], and in simulations [65] consistent with an increasingly exothermic signature of fibril elongation as the temperature increases. Interestingly, however, it was reported that the fibril elongation of $\alpha$-synuclein [35], as well as glucagon at moderate to high ionic strength [34] has a small positive heat capacity.

To probe the generality of our simulation results, we analysed isothermal titration calorimetry (ITC) measurements of the enthalpy of fibril elongation of several fibrils, both from our own experiments and from published work [33, 34], as well as calorimetric experiments of the dissolution of GNNQQNY crystals that we performed (see Materials and methods, S1 Text, S8 and S9 Figs). Furthermore, we included published calorimetric data of L-phenylalanine dissolution [60] and a van't Hoff analysis of di-phenylalanine crystallisation [61] (Fig 5A, S1 Table).

We find that in all cases the heat capacity of amyloid fibril elongation has negative values, see Fig 5A. In particular, also for glucagon and $\alpha$-synuclein. In the case of glucagon, we have performed the experiments under two different sets of solution conditions. Under the first

condition, 10 mM HCl and 1 mM $Na_2SO_4$, we find that $\Delta C_{p,el}$ (-2.0 kJ/mol) is in excellent agreement with previous reports [34] (-2.1 kJ/mol). In the second set of solution conditions (10 mM HCl and 30 mM NaCl), we find overall similar values of the heat of elongation compared to the first conditions and a slightly smaller, yet still negative, $\Delta C_{p,el}$ (-1.3 kJ/mol). This is surprising in the light of previous results that showed a positive value of $\Delta C_{p,el}$ in the presence of a higher NaCl concentration of 150 mM [34]. Note that higher salt concentrations would in general be expected to reinforce the strength of the hydrophobic effect. However, at high salt concentrations and elevated temperatures, the metastability of monomeric protein solutions can be reduced, which could result in the formation of aggregates in the protein solution that is used, affecting the ITC measurements.

For $\alpha$-synuclein we also find a negative $\Delta C_{p,el}$ (-3.2 kJ/mol) under very similar solution conditions (PBS buffer) to where it has previously been reported to have a positive value [35]. As these experiments are very sensitive to the exact state of the monomers and fibrils prior to titration, we performed the experiment in two different settings 1) by titrating fibrils into a solution of monomers and 2) by titrating a solution of monomers into seed fibrils, with consistent results, i.e. a negative $\Delta C_{p,el}$ (Fig 5A). We also performed calorimetric experiments on GNNQQNY crystals [66] (S7 Fig) which have so far not yet been thermodynamically characterized. We injected crystals into water and found in all cases endothermic signatures of crystal dissolution, corresponding to exothermic signatures of crystal growth (Fig 5A).

Therefore, our experimental results, together with the available data in the literature, suggest that for the large majority of experimentally accessible amyloid fibril systems, a negative value of $\Delta C_{p,el}$ is observed, in agreement with the predictions of both our theoretical and simulation models.

## Hydrophobicity dominates the thermodynamic signature of amyloid fibrils

One can wonder to what extent the hydrophobic temperature dependence dominates the observed thermodynamic signature for fibril elongation. Therefore, the question arises as to whether other non-covalent interactions may be responsible for the observation of a negative $\Delta C_{p,el}$ [67] in fibril elongation.

Firstly, it should be noted that mostly for small hydrophobic solutes strong negative heat capacities upon desolvation have been measured [60, 68] with additional evidence for phenylalanine and di-phenylalanine presented in Fig 5A. On the other hand, electrostatic interactions in solution [69] and hydrogen-bond forming substances show a weak temperature dependence for the solvation enthalpy. The temperature dependence of the hydrophobic effect is much more dramatic.

To test whether hydrophobicity can be the origin of the strong negative $\Delta C_{p,el}$ of fibril growth, we considered the total hydrophobic surface area per peptide buried upon assembly (amyloid fibril or crystal growth, see Materials and methods and S2 Methods) and associated this directly with the model for hydrophobic temperature dependence used in our simulations. Using Eq 11 we can analyse the differential enthalpy of amyloid fibril growth and peptide assembly with temperature. By fitting this equation to the experimental data of the temperature dependence of the enthalpies of assembly, we obtain an estimate of the strength of the hydrophobic effect ($\gamma$) for each of the assembling systems, see Fig 5A.

To determine the relationship between $\gamma$ and the buried hydrophobic surface area upon fibril growth, we calculated the total linear hydrophobic surface area of the peptides in the aggregating regions (see S3 Methods and S2 Table). As shown in the Fig 5B, the relationship between $\gamma$ and the aggregating hydrophobic surface area is linear, suggesting a direct dependence of the enthalpic signature on the hydrophobic surface area. Using this fit, we obtain a

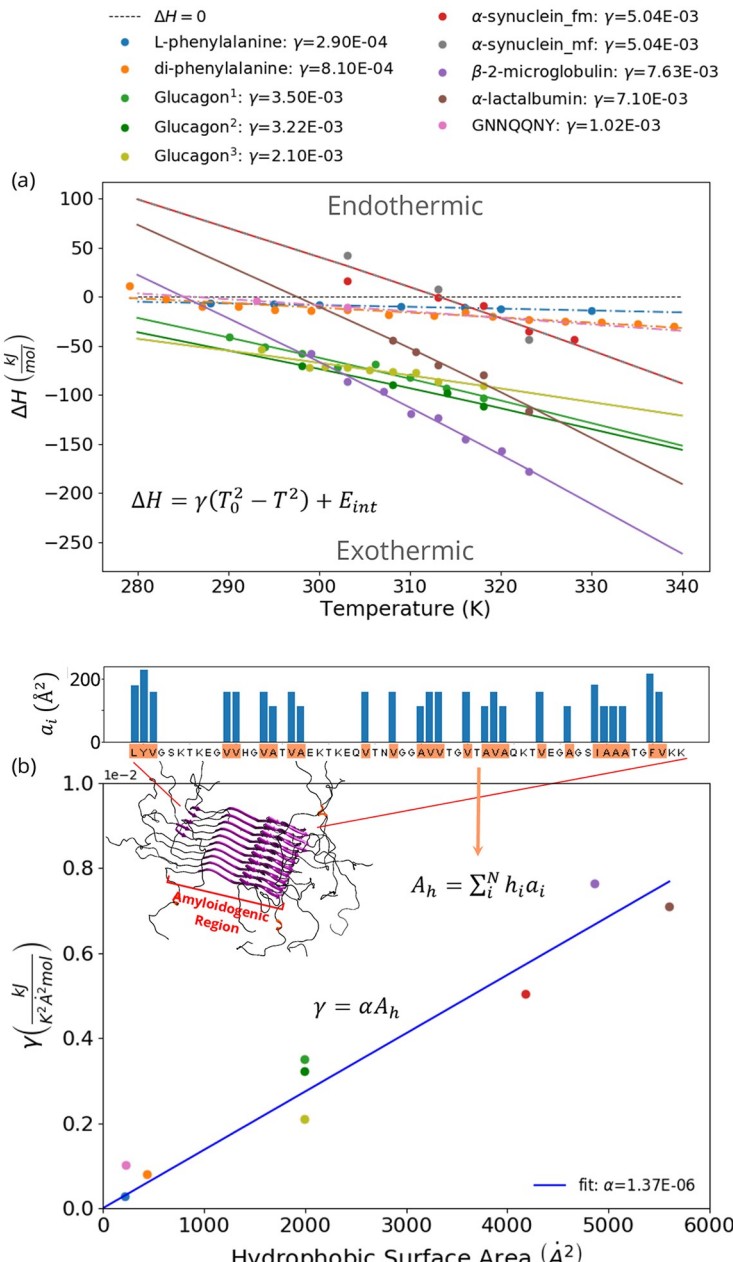

**Fig 5. Enthalpy of amyloid fibril elongation and peptide and amino acid crystallisation at different temperatures.**
The enthalpy of fibril elongation of $\alpha$-lactalbumin, $\alpha$-synuclein, glucagon and $\beta$-2-microglobulin [33] (solid lines), as well as the enthalpies of crystallisation of L-phenylalanine [60], diphenylalanine [61] and of GNNQQNY in water (dashed lines) are shown as a function of temperature. For $\alpha$-synuclein, ITC experiments were conducted in two set-ups: fibrils titrated into monomer solution (red symbols) and monomers titrated into fibril solution (grey symbols). See the Materials and methods and S2 Methods for experimental details. (A) Eq 11 was fitted through each of the curves to estimate the strength of the hydrophobic effect ($\gamma$) in $\frac{kJ}{K^2 mol}$. (B) Relationship between the fitted values of $\gamma$ and the total hydrophobic surface area of the proteins ($A_h$). [1]10 mM HCl, 1 mM $Na_2SO_4$, data from [34]. [2]10 mM HCl, 1 mM $Na_2SO_4$. [3]10 mM HCl, 30 mM NaCl.

strength of $1.42 \cdot 10^{-6} \frac{J}{\mathrm{mol}K^2A^{\circ 2}}$ for the hydrophobic effect per unit of hydrophobic surface area. Note that the fit through zero matches the experimental data, as is physically expected. These results strongly suggest that the general enthalpic signature of peptide assembly is directly associated with the hydrophobic temperature dependence. This conclusion is also fully compatible with previous results on the importance of the hydrophobic effect for the free energy barriers of amyloid fibril elongation [37], as well as with recent results from atomistic simulations that show that desolvation is the main driving force for amyloid fibril formation by the A$\beta$ peptide [70]. The importance of the hydrophobic effect for amyloid fibril stability is also likely to explain the finding that amyloid fibrils interact with lipids and biological membranes [71–73].

However, it is well-known that some amino acid sequences that are not usually classified as hydrophobic, such as poly-glutamine and GNNQQNY peptides can show $\beta$-strand dominated self-assembly. We show that even for the hydrophilic GNNQQNY peptide the hydrophobic signature for $\Delta C_{p,el}$ holds: as shown in Fig 5, the value of $\gamma$ for GNNQQNY is small, similar to that of di-phenylalanine, despite the substantial difference in molecular weight. As for the amyloid fibril formation by polyglutamine, it has been reported that monomeric polyglutamine peptides are considerably more compact in water compared to a fully denatured protein [74], i.e. that water is a bad solvent for polyglutamine. Therefore, the predictions of our model might have validity even beyond the systems that are traditionally classified as hydrophobic.

**The effect of salt on cold denaturation.**   Destabilisation of the fibril by charges [75] as a suggested prerequisite, but not necessarily cause, of cold denaturation of $\alpha$-synuclein [35], is therefore fully compatible with the results presented here. Fig 6 shows that in a solution at low ionic strength, $\alpha$-synuclein fibrils are less stable than in a solution at higher ionic strength. This can be explained by the stacking of monomeric units with identical charges in the fibrillar state, which repel each other, thereby destabilising the amyloid fibrils. The increase in the ionic strength of the solution can counter this effect by screening the repulsion [54]. Fig 6 also shows that in both cases (low and high ionic strength) the fibrils become less stable at low temperatures to a similar degree (cold-induced destabilisation of 12 kJ/mol at low ionic strength and of 8.5 kJ/mol at higher ionic strength). This result suggests that electrostatic effects modulate the absolute stability of amyloid fibrils, but do not drive cold denaturation of fibrils.

The modulation of absolute fibril stability by the solution conditions illustrated by the data in Fig 6 might also be able to provide an explanation for the qualitatively different temperature sensitivity reported for $\alpha$-synuclein fibrils, i.e. cold sensitive [35, 38] or cold-insensitive [76]: $\alpha$-synuclein fibrils are generally destabilised at low temperatures, but if the overall stability is too high, this destabilisation does not lead to an easily detectable increase in soluble concentration. Only under conditions of overall reduced stability, such as low ionic strength, cooling down will lead to measurable depolymerisation.

## Minimal requirements for cold denaturation

Combining the results from our theory, simulations, experiments and meta-data analysis, we show that there are three requirements for cold denaturation to occur in amyloid systems.

**(1) The hydrophobic contribution to the fibril stability must be a substantial fraction of the net free energy of fibril formation**.

In Fig 4B it can be observed that only if the hydrophobic temperature dependence is sufficiently strong (large $\alpha$) cold denaturation is observed. Moreover, in our model, fibrils simulated with moderate $\alpha$ values only destabilise at temperatures around or below the freezing point (corresponding to approximately T = 0.18 in reduced units). These findings are

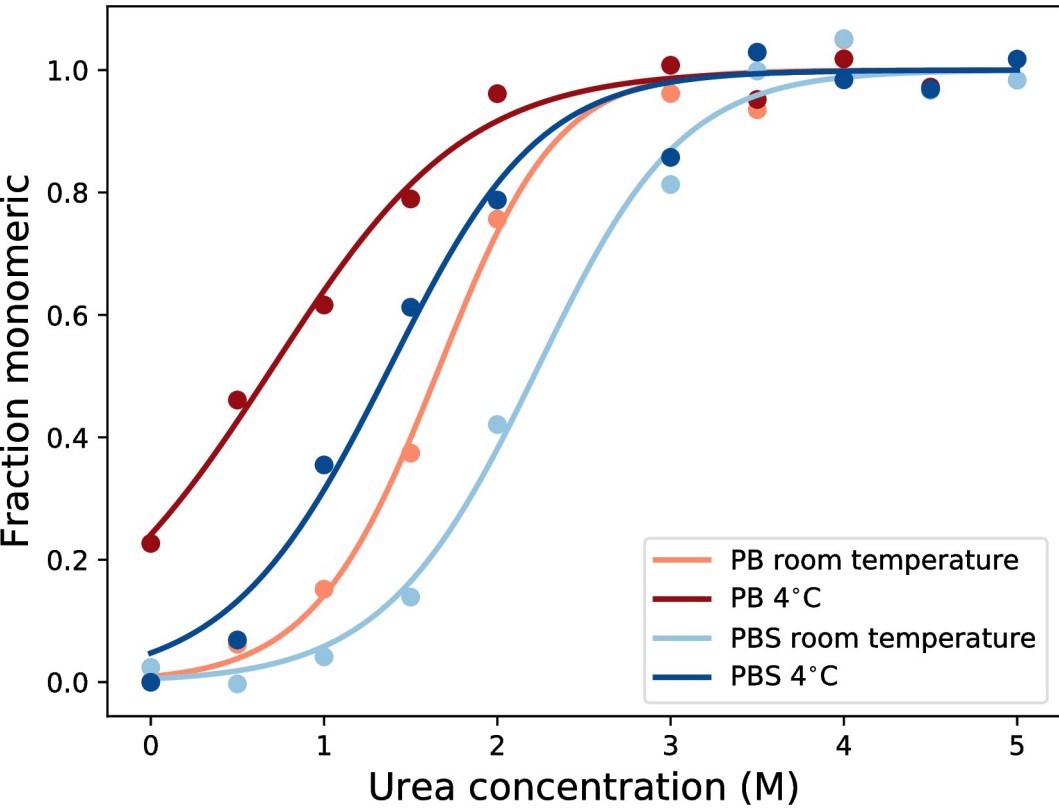

**Fig 6. The effect of changes in ionic strength and temperature on the stability of $\alpha$-synuclein fibrils.** We show the fraction of depolymerised fibrils as a function of urea concentration, determined by intrinsic fluorescence using the F94W mutant of $\alpha$-synuclein. Fibril destabilisation was measured at four different conditions: at high (dark, 10 mM PB + 150 mM NaCl) and low (light, 10 mM PB) ionic strength, and at 28˚C (red) and 4˚C. The depolymerisation curves were fitted with the isodesmic linear polymerisation model, Eq 12. The data shows that fibrils become less stable at low temperatures, and that an increase in ionic strength stabilises the fibrils by screening repulsive charges in the fibril.

supported by experimental measurements, showing that the elongation of the fibrils with the strongest hydrophobic core become endothermic at the highest temperatures (see Fig 5).

**(2) The overall stability of the fibril should be low to moderate**.

This is shown in Fig 4B, S2 and S3B Figs, where only fibrils with a moderate stability show denaturation at low temperatures. Note that a fibril can have low stability due to both entropic (configurational backbone entropy) and enthalpic (hydrogen-bonding, electrostatic repulsion) terms, see S2 and S3 Figs. This is also supported by the experimental results, which show that even if the influence of the hydrophobic effect is strong (steep slope in Fig 5a), the fibril elongation reaction remains exothermic at low temperatures if the fibril is very stable.

**(3) The enthalpy of the fibril elongation reaction needs to change sign from being exothermic at higher temperatures to endothermic at low temperatures**.

From Fig 3A, 3C and 5A we can observe that cold denaturation only occurs when the fibril elongation becomes endothermic at low temperatures. Note that only at temperatures below the transition from exothermic to endothermic the fibril fully destabilises.

The free energy difference between the fibrillar state and the denatured states contains large contributions from the configurational entropy of the polypeptide chain; therefore the exact transition temperatures for cold denaturation remain difficult to estimate for any given system without detailed simulations.

Many experimental observations can be explained by these requirements. Firstly, the strict requirements can explain why only a few proteins show cold denaturation of amyloid fibrils into monomers above the freezing point [35], whereas heat denaturation has been observed more generally [31, 32, 77]. More specifically, only $\alpha$-synuclein has been reported to display full cold denaturation, i.e. depolymerisation at low temperatures [35]. While all polypeptide systems that we have investigated display a negative $\Delta C_{p,el}$, we observe that the fibril growth reaction of $\alpha$-synuclein becomes endothermic at the highest temperature (Fig 5A). Moreover, for $\alpha$-synuclein the transition point from an exothermic to endothermic reaction occurs around 40˚C, while the onset of cold destabilisation is around 20˚C, leading to full cold denaturation close to the freezing point [35]. This is precisely as predicted by the simulations of models with a strong hydrophobic temperature dependence. In this context, it is also important to stress that amyloid fibrils of $\alpha$-synuclein have been found to be among the least stable in a large set of investigated polypeptides ([36], -33 kJ/mol in PBS), compatible with requirement 2. We obtain a very similar stability value for the F94W variant of $\alpha$-synuclein (-37.5 kJ/mol in PBS, Fig 6) that we have designed and produced in order to be able to use intrinsic fluorescence to conveniently follow fibril cold destabilisation and cold denaturation by intrinsic fluorescence [54]. In S10 Fig, we also show data with wild type $\alpha$-synuclein, using ThT fluorescence and CD spectroscopy that confirm the (partial) dissociation of the fibrils at low temperatures.

We have also performed chemical depolymerisation experiments (S6 Fig) of glucagon (-51.2 kJ/mol) and $\alpha$-lactalbumin amyloid fibrils (-52.5 kJ/mol) (S6 Fig) and we find that both types of amyloid fibrils are significantly more stable than the ones by $\alpha$-synuclein studied here.

The overall stability of amyloid fibrils corresponds to the net balance between the stabilising and destabilising factors. It has been proposed that electrostatic interactions are responsible for the cold denaturation of amyloid fibrils [35]. However, a destabilisation observed to act over the full temperature range is unlikely to originate from the hydrophobic effect, as shown by the simulations (S2 and S3 Figs). Furthermore, our equilibrium experiments show that electrostatic interactions are approximately equally destabilising at both low and moderate temperatures (Fig 6).

## Conclusion

In this work, we use a combination of theory, simulations, *in vitro* experiments and meta-data analysis to delineate the temperature dependencies of the stability of amyloid fibrils and their enthalpy of elongation. This approach highlights the importance of the interplay between the hydrophobic temperature dependence and the chain entropy and allows us to consider various enthalpy-entropy compensating mechanisms at a wide range of temperatures.

In summary, our results show that a large hydrophobic component of stability, low to moderate overall stability and a shift from exothermic to endothermic elongation of amyloid fibrils are necessary components to allow cold denaturation. Additionally, a strong temperature dependence of the enthalpy of fibril elongation is confirmed by ITC experiments, both from our own measurements and available data in the literature. Finally, the signature of the hydrophobic effect is visible in the negative $\Delta C_p$ of elongation for the large majority of investigated fibril systems. The magnitude of this negative heat capacity correlates closely with the hydrophobic surface area that is buried upon fibril or peptide crystal formation.

Hence, by delineating the necessary components for cold denaturation of amyloid fibrils, we can shed light on the crucial contribution of hydrophobicity to amyloid fibril stability and explain the observed enthalpic signature of amyloid fibril elongation. Moreover, it gives a direct way to estimate the total buried hydrophobic surface area via targeted ITC experiments.

## Supporting information

**S1 Methods. Details on the simulation methods and theory.**
(PDF)

**S2 Methods. Details on the experimental methods.**
(PDF)

**S3 Methods. Details on the meta data analysis methods.**
(PDF)

**S1 Text. Additional results to support the arguments in the main text.**
(PDF)

**S1 Table. Estimates of $\alpha$ from ITC experiments on four different proteins.** Data from the experiments was fitted to Eq 11 to estimate the value of $\alpha$.
(TSV)

**S2 Table. Sequences used to calculate the total hydrophobic surface area.** Only the amyloigenic regions of the fibrils were used (which does not equal the full sequence for $\alpha$-lactalbumin, $\alpha$-synuclein and $\beta$-2-microglobulin).
(TSV)

**S1 Fig. State diagrams for fibril elongation.** Here, we explored the effect of the strength of the hydrophobic temperature dependence, $\alpha$, on the stability of the aggregates. Three different states can be discerned: the fully formed fibril (black), denaturation of the fibril into monomers (cyan) and an amorphous aggregate where the two additional layers are not fully formed. Only in the models with a hydrophobic temperature dependence, cold destabilisation ($\alpha = 40$, 60), or cold denaturation into monomers, may be observed ($\alpha = 60$). The dashed lines indicate that the state has not been observed (sampled) in the simulations at the corresponding temperature. Note that the reduced temperature units for this model can be interpreted to have a freezing point around $T = 0.18$, and boiling point just above $T = 0.4$.
(TIF)

**S2 Fig. Varying the enthalpic contribution to fibril stability through hydrogen bonds.** We explored the stability of the fibrillar state for different values of the hydrogen bond strength ($\epsilon_{hb}$) in the model. For varying values $\epsilon_{hb}$, and $\alpha = 40$ the state diagram for the fibrillar state (A), the free energy (B), and corresponding entropic (C) and enthalpic (D) contributions are shown. Increasing the hydrogen bond strength makes the fibril more stable (b), resulting in a wider temperature range over which the fibrillar state is stable (a). Dotted lines indicate estimates for the hydrophobic contributions showing $\Delta \hat{G}_{\text{hydr}}$, $-T\Delta \hat{S}_{\text{hydr}}$ and $\Delta \hat{E}_{\text{hydr}}$; these estimates are generated using Eqns. 13, 15 and 14 with corresponding $\alpha$, $\Delta C_h = -6$ and with an offset, $E_{\text{int}} = \Delta H$ based on simulations with the equivalent peptide for $\alpha = 0$.
(TIF)

**S3 Fig. Varying the entropic contribution to fibril stability through chain entropy.** We explored the stability of the fibrillar state for different values of the (entropic) propensity of $\beta$-strand state ($N_\beta$) in the model. For varying values $N_\beta$, and $\alpha = 40$ the state diagram for the fibrillar state (A), the free energy (B), and corresponding entropic (C) and enthalpic (D) contributions are shown. Increasing the $\beta$-strand propensity makes the fibril more stable (B), resulting in a wider temperature range over which the fibrillar state is stable (A). Dotted lines indicate estimates for the hydrophobic contributions showing $\Delta \hat{G}_{\text{hydr}}$, $-T\Delta \hat{S}_{\text{hydr}}$ and $\Delta \hat{E}_{\text{hydr}}$;

these estimates are generated using Eqns. 13, 15 and 14 with corresponding $\alpha$, $\Delta C_h = -6$ and with an offset, $E_{\text{int}} = \Delta H$ based on simulations with the equivalent peptide for $\alpha = 0$.
(TIF)

**S4 Fig. Representative ITC raw data.** Raw data of ITC experiments are shown for experiments where monomer solutions were titrated into seed fibril suspensions. Experiments were performed with a VP-ITC (A,C) and an ITC200 (C) instruments. (A) Injections of 10, 80, 80, 80$\mu$l (40˚) and 10, 80, 40, 80$\mu$l (50˚) of a solution of $\alpha$-lactalbumin (50 $\mu$M in 10 mM HCl +100mM NaCl) into a suspension of sonicated seed fibrils. (B) Injections of 2 $\mu$l of solutions of monomeric $\alpha$-synuclein at 380 $\mu$M (50˚C), 390 $\mu$M (30˚C) and 430 $\mu$M (40˚C) into seed fibril suspensions. (C) Injections of 20, 80, 80, 80$\mu$l (30 and 47˚) of a solution of glucagon (100 $\mu$M in 10 mM HCl+30mM NaCl) into a suspension of sonicated seed fibrils.
(TIF)

**S5 Fig. AFM images illustrating $\alpha$-synuclein amyloid fibril elongation.** Atomic force microscopy (AFM) images were taken of seed fibrils before sonication to shorten the length distribution and enhance the seeding efficiency (A), after 10 s of sonication with a sonication probe (B) and after an ITC experiment (C), where the fibrils (40 $\mu$M) had been incubated with a total of 60 $\mu$M of monomeric $\alpha$-synuclein.
(TIF)

**S6 Fig. Chemical depolymerisation of amyloid fibrils.** (A) $\alpha$-lactalbumin amyloid fibrils depolymerised with the strong denaturant GndSCN. (B) glucagon amyloid fibrils depolymerised with GndHCl. The values of the free energy difference between the soluble and fibrillar states are -52.5 kJ/mol ($\alpha$-lactalbumin) and -51.2 kJ/mol (glucagon). These values should be compared with the one determined for the considerably less stable $\alpha$-synuclein amyloid fibrils of -33.0 kJ/mol [36] or -37.4kJ/mol (this study) in PBS buffer.
(TIF)

**S7 Fig. Thermodynamics of GNNQQNY crystallisation.** (A) Raw ITC data of the injectionof GNNQQNY crystals and monomer into pure water. Experimental details see the Materials and methods section. (B) Summary of the calorimetric results of GNNQQNY crystal dissolution. The data points are corrected for the exothermic heats of dilution of the monomeric content of each injection. The linear fits to the data sets at the two temperatures are used to determine the molar enthalpies of crystal dissolution, which corresponds to the negative of the molar heats of crystal growth.
(TIF)

**S8 Fig. Intrinsic fluorescence spectra of the F94W mutant of $\alpha$-synuclein.** Left: Intrinsic fluorescence of monomeric F94W $\alpha$-synuclein was measured in PB pH 7.4 in the presence and absence of 150 mM NaCl and in the presence and absence of 5 M urea. Right: The fluorescence intensity ratios at 340 and 330 nm were plotted for all 4 conditions, confirming that this ratio is nearly constant for monomer under all the observed conditions. These results provide strong support for our interpretation of the change in fluorescence intensity ratio with increasing urea concentration as reflecting fibril depolymerization.
(TIF)

**S9 Fig. Equilibration of $\alpha$-synuclein fibrils at 4˚C.** Urea denaturation series of F94W $\alpha$-synuclein fibrils in the presence and absence of 150 mM NaCl were incubated at 4˚C for 4 weeks and the intrinsic fluorescence spectra were measured and the overall denaturation curves

determined and analysed with the isodesmic polymerisation model. The determined free energy values were plotted as a function of equilibration time. It can be seen that equilibrium is reached after a few days.
(TIF)

**S10 Fig. Cold denaturation of WT $\alpha$-synuclein amyloid fibrils followed by ThT fluorescence and CD spectroscopy.** (A) $\alpha$-synuclein amyloid fibril stability as a function of urea concentration at room temperature and after equilibration at 7˚C, using Thioflavin-T fluorescence as a read-out for the degree of fibril depolymerisation. (B) Time course of partial fibril depolymerisation at 7˚C, as followed by circular dichroism (CD) spectroscopy. See the Materials and methods section for experimental details. The data have been fitted to negative exponential functions, corresponding to the expected behaviour of fibril dissolution, whereby the number of fibrils is approximately constant during most of the dissociation time course and where the rate of dissociation is proportional to the difference between the equilibrium concentration and the actual concentration of monomer.
(TIF)

## Acknowledgments

We thank Alexandra Ziemski for help with the production of $\alpha$-synuclein and Sara Linse for help with the Thioflavin-T cold denaturation experiments.

## Author Contributions

**Conceptualization:** Erik van Dijk, Alexander K. Buell, Sanne Abeln.

**Data curation:** Juami Hermine Mariama van Gils, Alessia Peduzzo.

**Formal analysis:** Juami Hermine Mariama van Gils, Erik van Dijk, Alessia Peduzzo, Halima Mouhib, Alexander K. Buell.

**Funding acquisition:** Georg Groth, Halima Mouhib, Alexander K. Buell, Sanne Abeln.

**Investigation:** Juami Hermine Mariama van Gils, Erik van Dijk, Alessia Peduzzo, Alexander Hofmann, Nicola Vettore, Marie P. Schützmann, Halima Mouhib, Daniel E. Otzen, Alexander K. Buell, Sanne Abeln.

**Methodology:** Juami Hermine Mariama van Gils, Erik van Dijk, Alessia Peduzzo, Alexander Hofmann, Nicola Vettore, Marie P. Schützmann, Georg Groth, Halima Mouhib, Daniel E. Otzen, Alexander K. Buell, Sanne Abeln.

**Project administration:** Alexander K. Buell, Sanne Abeln.

**Software:** Juami Hermine Mariama van Gils, Erik van Dijk, Halima Mouhib, Sanne Abeln.

**Supervision:** Daniel E. Otzen, Alexander K. Buell, Sanne Abeln.

**Validation:** Alessia Peduzzo, Alexander Hofmann, Nicola Vettore, Marie P. Schützmann, Georg Groth, Daniel E. Otzen, Alexander K. Buell.

**Visualization:** Juami Hermine Mariama van Gils, Erik van Dijk, Alessia Peduzzo, Alexander K. Buell, Sanne Abeln.

**Writing – original draft:** Juami Hermine Mariama van Gils, Erik van Dijk, Alessia Peduzzo, Halima Mouhib, Alexander K. Buell, Sanne Abeln.

**Writing – review & editing:** Juami Hermine Mariama van Gils, Erik van Dijk, Alessia Peduzzo, Alexander Hofmann, Nicola Vettore, Marie P. Schützmann, Georg Groth, Halima Mouhib, Daniel E. Otzen, Alexander K. Buell, Sanne Abeln.

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
