## [Decision Letter · Decision Letter 0]

9 Feb 2020

Dear Dr. Abeln,

Thank you very much for submitting your manuscript "The hydrophobic effect characterises the thermodynamic signature of amyloid fibril growth" for consideration at PLOS Computational Biology. As with all papers reviewed by the journal, your manuscript was reviewed by members of the editorial board and by several independent reviewers. The reviewers appreciated the attention to an important topic. Based on the reviews, we are likely to accept this manuscript for publication, providing that you modify the manuscript according to the review recommendations.

Sincerely,

Nikolay V. Dokholyan, PhD

Guest Editor

PLOS Computational Biology

Nir Ben-Tal

Deputy Editor

PLOS Computational Biology

[LINK]

Reviewer's Responses to Questions

**Comments to the Authors:**

Reviewer #1: The authors combined CG simulations, in vitro measurements, and meta-analysis to study the thermodynamics of fibril growth/elongation. With CG simulations, the authors illustrated several requirements for the observation of cold denaturation of fibrils and showed that amyloid fibril growth/elongation is associated with a negative heat capacity. They also performed in vitro experiments to support their results. Overall, the results are interesting and the study deserves publication. Here are a few minor issues:

1. For the CG model, T0 value (the temperature-dependent terms of protein-solvent interaction) was not given.

2. For the depolymerisation experiments and cold-denaturation of fibrils (e.g., Fig. 6), it would be great if additional approaches could be used to cross-validate the cold-denaturation, such as AFM or TEM imaging.

Reviewer #2: The manuscript attempts to obtain insight into fibril formation, using combination of experimental, theoretical and simulation techniques. From simulation viewpoints the authors use improved cubic lattice protein model which can account for hydrogen bonds and hydrophobic interactions. The author simulate fibril elongation and perform ITC experiments on elongation and study cold denaturation. The author obtain qualitative agreement between simulations and experiment. They obtain insight on importance of hydrophobicity for the process.

This looks an interesting computational and experimental study which fits the scope of Plos Comp bio and could be published

**Have all data underlying the figures and results presented in the manuscript been provided?**

Reviewer #1: Yes

Reviewer #2: Yes

PLOS authors have the option to publish the peer review history of their article (what does this mean?). If published, this will include your full peer review and any attached files.

Reviewer #1: No

Reviewer #2: No
---

## [Editor Report · Decision Letter 1]

2 Mar 2020

Dear Dr. Abeln,

We are pleased to inform you that your manuscript 'The hydrophobic effect characterises the thermodynamic signature of amyloid fibril growth' has been provisionally accepted for publication in PLOS Computational Biology.

Best regards,

Nikolay V. Dokholyan, PhD

Guest Editor

PLOS Computational Biology

Nir Ben-Tal

Deputy Editor

PLOS Computational Biology

---

## [Editor Report · Acceptance letter]

24 Apr 2020

PCOMPBIOL-D-19-02211R1 

The hydrophobic effect characterises the thermodynamic signature of amyloid fibril growth

Dear Dr Abeln,

I am pleased to inform you that your manuscript has been formally accepted for publication in PLOS Computational Biology. Your manuscript is now with our production department and you will be notified of the publication date in due course.

With kind regards,

Sarah Hammond
